# The Application of ATR-FTIR Spectroscopy and the Reversible DNA Conformation as a Sensor to Test the Effectiveness of Platinum(II) Anticancer Drugs

**DOI:** 10.3390/s18124297

**Published:** 2018-12-06

**Authors:** Khansa Al-Jorani, Anja Rüther, Miguela Martin, Rukshani Haputhanthri, Glen B. Deacon, Hsiu Lin Li, Bayden R. Wood

**Affiliations:** 1Centre for Biospectroscopy and School of Chemistry, Monash University, Clayton, VIC 3800, Australia; khansa.al-jorani@monash.edu (K.A.-J.); anja.ruether@monash.edu (A.R.); miguela.martin@monash.edu (M.M.); rukshaniH@slintec.lk (R.H.); 2School of Chemistry, Monash University, Clayton, VIC 3800, Australia; glen.deacon@monash.edu (G.B.D.); hsiu.li@unsw.edu.au (H.L.L.); 3Presently at School of Chemistry, UNSW Sydney, Sydney, NSW 2052, Australia

**Keywords:** platinum, DNA, IR

## Abstract

Platinum(II) complexes have been found to be effective against cancer cells. Cisplatin curbs cell replication by interacting with the deoxyribonucleic acid (DNA), reducing cell proliferation and eventually leading to cell death. In order to investigate the ability of platinum complexes to affect cancer cells, two examples from the class of polyfluorophenylorganoamidoplatinum(II) complexes were synthesised and tested on isolated DNA. The two compounds *trans*-[*N*,*N*′-bis(2,3,5,6-tetrafluorophenyl)ethane-1,2-diaminato(1-)](2,3,4,5,6-pentafluorobenzoato)(pyridine)platinum(II) (PFB) and *trans*-[*N*,*N*′-bis(2,3,5,6-tetrafluorophenyl)ethane-1,2-diaminato(1-)](2,4,6-trimethylbenzoato)(pyridine)platinum(II) (TMB) were compared with cisplatin through their reaction with DNA. Attenuated Total Reflection Fourier Transform Infrared (ATR-FTIR) spectroscopy was applied to analyse the interaction of the Pt^(II)^ complexes with DNA in the hydrated, dehydrated and rehydrated states. These were compared with control DNA in acetone/water (PFB, TMB) and isotonic saline (cisplatin) under the same conditions. Principle Component Analysis (PCA) was applied to compare the ATR-FTIR spectra of the untreated control DNA with spectra of PFB and TMB treated DNA samples. Disruptions in the conformation of DNA treated with the Pt^(II)^ complexes upon rehydration were mainly observed by monitoring the position of the IR-band around 1711 cm^−1^ assigned to the DNA base-stacking vibration. Furthermore, other intensity changes in the phosphodiester bands of DNA at ~1234 cm^−1^ and 1225 cm^−1^ and shifts in the dianionic phosphodiester vibration at 966 cm^−1^ were observed. The isolated double stranded DNA (dsDNA) or single stranded DNA (ssDNA) showed different structural changes when incubated with the studied compounds. PCA confirmed PFB had the most dramatic effect by denaturing both dsDNA and ssDNA. Both compounds, along with cisplatin, induced changes in DNA bands at 1711, 1088, 1051 and 966 cm^−1^ indicative of DNA conformation changes. The ability to monitor conformational change with infrared spectroscopy paves the way for a sensor to screen for new anticancer therapeutic agents.

## 1. Introduction

The Pt^(II)^ complexes, carboplatin, cisplatin and oxaliplatin have been approved worldwide for the treatment of cancerous tumours [1,2,3,4]. Cisplatin was approved as an anticancer drug by the U.S Food and Drug Administration in 1978 [5]. Soon after it became a wide spectrum anticancer agent because of its activity against non-small lung cancer, head, breast and neck cancers [6,7], in addition to its specific use in testicular and ovarian cancer. The interaction of cisplatin with deoxyribonucleic acid (DNA) inside the nucleus causes cell death and apoptosis. Previous studies have found that the interaction occurs between cisplatin and the DNA bases, especially guanine-N(7) but also adenine-N(3) [8]. The interaction of cisplatin with DNA has been studied extensively using X-ray crystallography and NMR spectroscopy in addition to Liquid Chromatography Mass Spectrometry (LCMS) [5,9,10]. Resonant inelastic X-ray scattering (RIXES) spectroscopy has mapped the hydration of cisplatin and its binding to adjacent guanine bases of DNA [11]. The cisplatin-resistance of some cancer types and dose-limiting side effects such as nephrotoxicity, peripheral neuropathy, vomiting, renal and visual impairment, restrict its application [12]. These effects have led to a search for novel Pt^(II)^ complexes with increased stability that destroy cancer cells with less side effects [13]. The side effects are postulated to result from non-selective reactions of cisplatin with other biomolecules such as proteins and phospholipids as well as interactions of the drug with healthy tissues leading to dose limiting nephrotoxicity [13,14].

Because the main target of the platinum complexes is DNA, understanding the effect of these drugs on DNA conformation is essential [8,15]. After hydrolysis of the chloride leaving group the complex is in a form in which it can interact with DNA [11,16,17]. It was found that cisplatin binds to double stranded (ds) DNA through coordination by the guanine base at the N7 position, leading to changes in the DNA structure [11,16,18]. Studies performed on DNA using X-ray crystallography showed that the interaction of cisplatin with DNA causes the DNA helix to unwind and distort [11,18]. The interaction between the DNA and the drug can be studied with infrared (IR) spectroscopy because the method is very sensitive towards DNA conformational changes. Thus, the technique provides a useful tool to explore the effects of drugs on the conformation of this fundamentally important biological molecule [19]. 

“Rule breakers” which do not conform to the usual structure/activity rules for platinum(II) complexes [20,21,22] represent an approach to overcome the current deficiencies of clinical drugs. One such class is platinum(II) organoamides [Pt(NRCH_2_)_2_(py)_2_] (R = polyfluorophenyl; py = pyridine), of which [Pt{N(*p*-HC_6_F_4_)CH_2_}_2_(py)_2_] (Pt103) is the lead compound. It is active in vitro against both cisplatin active and resistant cells and against some in vivo tumours [23] and has been shown to have greater cellular uptake and a large number of DNA inter-strand cross links compared with cisplatin [24]. Recently, RIXES spectroscopy demonstrated that Pt103 initially reacts preferentially with adenine bases of DNA [25], thus offering an explanation for the difference in biological properties from cisplatin. Pt103 is unusual as an anticancer active molecule with four nitrogen donor atoms and is a “rule breaker” because of no N-H bonds on the nitrogen donor atoms.

We have recently shown by FTIR methods [26] that Pt103 interacts with DNA in aqueous acetone prior to hydrolysis. Such an interaction presumably involves H-bonding. 

In an attempt to provide more opportunity for H-bonding with DNA, we reacted Pt103 with pentafluorobenzoic acid and 2,4,6-trimenthylbenzoic acid giving the two compounds namely *trans*-[*N*,*N*′-bis(2,3,5,6-tetrafluorophenyl)ethane-1,2-diaminato(1-)](2,3,4,5,6-pentafluorobenzoato)(pyridine)platinum(II).

(**PFB**) and *trans*-[*N*,*N*′-bis(2,3,5,6-tetrafluorophenyl)ethane-1,2-diaminato(1-)](2,4,6-trimethylbenzoato)(pyridine)platinum(II) (**TMB**) (Figure 1). **PFB** and **TMB**, contain N-H bonds (consistent with structure-activity rules) and therefore should lead to enhanced H-bonding with DNA. We now report an investigation of the interaction of **PFB** and **TMB** with DNA by ATR-FTIR spectroscopy.

In examining how these Pt^(II)^ complexes affect the DNA conformation we performed a series of ATR-FTIR studies monitoring hydrated, dehydrated and rehydrated DNA-drug combinations compared to untreated control DNA in solvent mixtures.

Watson and Crick originally detected the polymorphism of the DNA 60 years ago. Later, Franklin and Gosling indicated that the DNA forms two different conformations, namely the hydrated B-DNA form and the dehydrated A-DNA form [27]. 

The DNA conformation is also sensitive towards pH, temperature, counter ion and base pair sequences [28], which were kept constant in this study. In the absence of drugs that interact with DNA, the DNA conformation reversibly goes from B-DNA in the hydrated state to A-DNA in the dehydrated state and back to B-DNA upon rehydration. It was hypothesised that the effect of Pt^(II)^ compounds could be investigated by studying the dynamical conformation change from the A-DNA form to the B-DNA form in the presence and absence of Pt^(II)^ novel complexes upon rehydration (Figure 2). Hydration and electrostatic interactions are considered major factors leading to the transition. Disruption of these electrostatic interactions can be monitored by recording FTIR spectra of DNA before and after rehydration.

The drugs were mixed with calf thymus dsDNA and ATR-FTIR spectra were measured to investigate changes in FTIR bands that are concomitant with changes in the DNA conformation. Repeated measurements of FTIR spectra were performed to obtain a series of spectra of the dehydrated samples, starting with the hydrated sample, which was deposited in an aqueous state onto the ATR crystal and left until the water stretching mode at ~3650 cm^−1^ disappeared. Then each sample was exposed to the gradual addition of water, which initiated the process of rehydration and was monitored by ATR-FTIR spectroscopy.

## 2. Materials and Methods

### 2.1. Biological Compounds

Double stranded calf thymus DNA (mol. Wt. 8–15 kb; 42% GC content) (Sigma, St. Louis, MO, USA), single stranded calf thymus DNA (Sigma, St. louis, MO, USA), acetone (Merck KGaA, Darmastadt, Germany), Tris-HCl buffer (2-amino-2-hydroxymethyl-propane-1,3-diol hydrochloride) (Merck KGaA, Darmstadt, Germany), hydrochloric acid (HCl) 32% (Ajax Fine Chemicals Pty Ltd., Taren Point, NSW, Australia). The Pt103 was prepared by the reported method [29].

### 2.2. Platinum Compounds Synthesis

The platinum complexes **PFB** and **TMB** were synthesized as shown in Figure 1.

#### 2.2.1. Synthesis of **PFB**

[*N*,*N*′-Bis(2,3,5,6-tetrafluorophenyl)ethane-1,2-diaminato(1-)](pentafluorobenzoato)(pyridine)platinum(II) (**PFB**).

[*N*,*N*′-Bis(2,3,5,6-tetrafluorophenyl)ethane-1,2-diaminato(2-)]dipyridineplatinum(II)) (Pt103) (0.20 g, 0.28 mmol) was treated with) pentafluorobenzoic acid (0.060 g, 0.28 mmol) in diethyl ether (160 mL) and was stirred and irradiated by light (halogen lamp 500 W) for 240 min. The crude product was isolated by evaporation to dryness and dissolved in a minimum amount of acetone and passed through a 15 cm neutral alumina column with ethyl acetate: light petroleum: acetone 1:1.5:0.1 as eluent. The yellow band indicative of the product was collected and the solution evaporated to dryness. Diethyl ether ~2 mL and petroleum ether ~5 mL were added and cooled. The compound produced bright yellow crystals. Yield 0.090 g, 38%, m.p. 174 °C (decomposed). (Found: C, 37.5; H, 1.3; N, 5.1. C_26_H_12_F_13_N_3_O_2_Pt requires C, 37.2; H, 1.4; N, 5.0%). ^1^HNMR: 2.92–3.19, br m, 3H, C**H**_2_NH and C**H**_2_N; 4.07, m, 1H, C**H**_2_NH; 6.36, m, 1H, *p*-**H**C_6_F_4_N; 7.36, t, 2H, H3, 5 (py); 7.54, m, 1H, *p*-**H**C_6_F_4_NH; 7.91, t, 1H, H4(py); 8.35, br, ˂1H, N**H**; 8.63, d with ^195^Pt satellites, ^3^J_H,H_ 5 Hz ^3^J_H,Pt_ 39 Hz, 2H, **H**2,6(py). The broad NH resonance integrated less than the expected 1H (Appendix A). ^19^FNMR.: −140.8, m, 2F, **F**3,5(*p*-HC_6_F_4_NH); −144.85, m, 2F, **F**2,6(O_2_CC_6_F_5_); −145.1, m, 2F, **F**3,5(*p*-HC_6_F_4_N); −146.9, br m, 2F, **F**2,6(*p*-HC_6_F_4_NH); −153.3, m, 2F, **F**2,6(*p*-HC_6_F_4_N); −158.4, t, ^3^J_F,F_ 20 Hz, 1F, **F**4(O_2_CC_6_F_5_), −164.5, m, 2F,**F**3,5(O_2_CC_6_F_5_). UV/Visible spectrum λ max (ε): 390 nm (2.2 × 10^3^) (Appendix A). IR spectrum: 3104m, 3075sh, 2962w, 2868w, 2647w, 2324w, 2113w, 1982w, 1920w, 1659vs, 1631vs, 1583w, 1528s, 1493s, 1473s, 1452s, 1427w, 1414w, 1405w, 1361vs, 1336m, 1285w, 1260s, 1216w, 1180m, 1158m, 1133m, 1101m, 1082s, 1047w, 1019s, 990vs,935m, 897m, 869m, 843w, 819w, 799w, 786vs,758m, 716s, 694w, 667w, 641w cm^−1^ (Appendix A) Mass spectrum (ESMS + ve): *m*/*z* 863 [15, (M + Na)^+^]; 841 [100, (M + H)^+^]; 763 [5, (M-C_5_H_3_N)^+^]; 661 [15, (M-C_6_F_5_CO_2_ + MeOH)^+^]; 629 [30, (M-C_6_F_5_CO_2_)^+^] (Appendix A).

#### 2.2.2. Synthesis of **TMB**

[*N*,*N*′-Bis(2,3,5,6-tetrafluorophenyl)ethane-1,2diaminato(1-)](pyridine)(2,4,6-trimethylbenzoato)platinum(II) (**TMB**).

Pt 103 (0.20 g, 0.28 mmol) and 2,4,6-trimethylbenzoic acid (0.050 g, 0.28 mmol) were stirred together in diethyl ether (160 mL) and irradiated by light (halogen lamp 500W) for 240 min. Evaporation, dissolution in acetone and chromatography were performed as for **PFB** above. Diethyl ether and petroleum ether were added to the residue and cooled. The compound gave bright yellow crystals. Yield 0.080 g, 36%, m.p. 165 °C (dec.). (Found: C, 44.0; H, 3.1; N, 5.5. C_29_H_23_F_8_N_3_O_2_Pt requires C,43.9;H,2.9;N,5.3%). 1H NMR: 1.57, s, 6H, *o*-CH_3_; 2.10, s, 3H, *p*-CH3; 2.96–3.25, brm, 3H, C**H**_2_N and C**H**_2_NH; 4.14, m, 1H, C**H**_2_NH; 6.24, m, 1H, *p*-**H**C_6_F_4_N; 6.55, s, 2H, H3,5(Ph); 7.33, t, 3JH,H 7Hz, 2H, H3,5(py); 7.65, m, 1H, *p*-**H**C_6_F_4_NH; 7.88, t, ^3^JH, H 8Hz, 1H, H4(py); 8.64, d with ^195^Pt satellites, ^3^J_H,H_ 5 Hz, ^3^J_H,Pt_ 38 Hz, ˂3H, **H**2,6(py) and N**H**. This feature integrated for less than the expected 3H (Appendix A). ^19^FNMR: −140.7, m, 2F, **F**3,5; (*p*-HC_6_F_4_NH); −145.3, m, 2F, **F**3,5 (*p-*HC_6_F_4_N); −146.1, m, 2F, F2,6 (*p*-HC_6_F_4_NH); −154.1, m, 2F, F2,6 (*p*-HC_6_F_4_N). UV/Visible spectrum λ max (ε): 376nm (2.2 × 10^3^) (Appendix A). IR spectrum: 3454w, 3080mw, 3048w, 3032w, 3010w, 2982w, 2943w, 2862w, 2323w, 2167w, 2113w, 1981w, 1916w, 1739vw, 1632w, 1608vs, 1581w, 1523vs, 1492s, 1467mw, 1453s, 1443s, 1351vs, 1288mw, 1266m, 1245w, 1213w, 1177m, 1157m, 1123s, 1076m, 1044s, 1019w, 936vw, 895vs, 869m, 856w, 841w, 831m, 791m, 767m, 743w, 716mw, 696s, 674w, 663w, 629mw cm^−1^ (Appendix A). Mass spectra: *m*/*z* 815 [15, (M + Na)^+^]; 793 [100, (M + H)^+^]; 714 [20, (MH-py)^+^]; 661 [15, (M-C_6_H_2_Me_3_CO_2_ + MeOH)^+^]; 629 [25, (M-C_6_H_2_Me_3_CO_2_)^+^] (Appendix A).

### 2.3. Preparation of DNA Solutions

Double stranded calf thymus DNA (1 mg/mL) and single stranded calf thymus DNA (0.75 mg/mL) solutions were obtained from Sigma Chemical Company and prepared in 10 mM Tris-HCl buffer at pH 7.4 and stored at 8 °C. The stock solutions of 6 mM and 3 mM of both **PFB** and **TMB** were prepared in water/acetone (10% *v*/*v*) mixture and stored at room temperature in the dark. The colour of the fresh solution was bright yellow. These solutions remained stable for a few months. UV-Vis spectra and cyclic voltagrams were acquired to confirm the stability of these drug solutions.

### 2.4. Preparation of Drug Solution

The two compounds **PFB** and **TMB** were prepared as stock solutions of 6.00 mM in water: acetone (10% *v*/*v*).

Stock solution calf thymus dsDNA or ssDNA (100 μL) were added to 100 μL of the stock solution of the Pt drug solution at room temperature and mixed for 2 min (Vortexer, Ratek Instruments Pty Ltd., Knox City, Victoria, Australia). It was wrapped tightly using parafilm to prevent liquid evaporation and incubated for 48 h at 37 °C in the dark. 3 μL aliquots of the clear solution were collected and deposited onto the ATR crystal. All experiments were carried out in triplicates. The average observed pH of the final mixtures of DNA treated with Pt drug solution or DNA control was 7.2 (±0.04). The pH value was measured with a pH meter (Hanna Instruments Pty. Ltd., Woonsocket, RI, USA).

### 2.5. Instrumentation

UV-Vis spectra were acquired using a Carry 100 UV-Vis spectrometer with the Varian Carry WinUV software (Santa Clara, CA, USA) for both double strand and single strand DNA. Both solutions showed a UV band at 260 nm [30].

ATR-FTIR spectra of DNA were acquired using a Silicon BioATR Cell II accessory (Harrick Scientific, Pleasantville, NY, USA) coupled to a Bruker IFS Equinox55 FTIR system (Bruker Optics Pvt. Ltd., Billerica, MA, USA). The silicon ATR crystal has an inert sample interface (Teflon and stainless steel) and *ca* 6 μm effective pathlength between 1500 to 2000 cm^−1^. ATR-FTIR spectra of solid drugs were recorded using a Golden Gate single bounce diamond micro-ATR coupled to a Bruker IFS Equinox FTIR system (Bruker Optics Pvt. Ltd., Billerica, MA, USA). The diamond ATR crystal has *ca* 2 μm effective pathlength at 1000 cm^−1^ and was used to record ATR-FTIR spectra for platinum compounds in the solid state. The data were processed using the Bruker OPUS software, version 6.0 (Bruker Optics Pvt. Ltd.). 

### 2.6. UV-Vis Spectra of DNA

The stock solution of dsDNA calf thymus was prepared at the concentration of 1 mg/mL. The UV-Vis absorbance was measured for diluted DNA solution (50× dilution). The acquisition of UV-Vis absorbance was achieved using a 5 mL quartz cell and a Carry UV-Vis spectrometer at 260 nm. The purpose of this step is to extract the nucleotide concentration by calculating the PO_2_^−^ concentration as they are equivalent to each other. 

### 2.7. ATR-FTIR Spectroscopy

The spectra of the solid drugs were recorded using the Golden Gate single bounce diamond micro-ATR system coupled to a Bruker IFS Equinox FTIR system (Bruker). After cleaning the surface of the ATR crystal with distilled water and isopropanol, the samples of DNA-drug solutions were recorded using the silicon ATR crystal (45 °C top plate) of the BioATRCell II, which has an inert sample interface (Teflon and stainless steel) and *ca* 6 μm effective pathlength between 1500–2000 cm^−1^.

Three replicates were recorded for each drug-DNA sample and control. The samples included the DNA solution with Tris buffer solution, a DNA solution with a solvent (acetone:water mixture 10:1) and DNA samples mixed with drug solutions. 3 μL of each aqueous sample was placed onto the silicon ATR crystal of the BioATR cell covering the entire crystal to ensure the coverage of the 4.4 mm diameter of the crystal surface and to provide an active sampling area by forming a uniform film. In the spectral region from 4000 to 600 cm^−1^, 50 sample interferograms were acquired at a resolution of 4 cm^−1^ with a zero filling of 2. The spectrum of blank silicon was acquired as the background before each sample spectrum. Before transferring the samples to the ATR crystal, they were left on the bench for a few minutes to establish room temperature. The spectra of each sample were acquired continuously over a one hour period every 60 s. The samples included the DNA solution mixed with saline or water/acetone as the controls and the DNA solutions treated with **PFB**, **TMB** and cisplatin, respectively. The drug was dissolved in an acetone water mixture (10:1) and incubated for 48 h at the physiological temperature of 37 °C to mimic the body temperature. 3 μL of each sample was deposited on the ATR biocell and air-dried until consistent spectra were achieved. The process was repeated to obtain a series of dehydrated samples.

The spectra showing acetone contamination observed in the first two or three spectra for each sample were excluded from the ensuing analysis. 

The rehydration procedure utilised a humidifier to apply a stream of mist over each dehydrated sample. The rehydration process was monitored for each sample with 60 scans at 8 cm^−1^ resolution until the DNA control and DNA: drug samples were fully hydrated. The rehydration process was monitored by the increasing intensity of the OH stretching mode of water around 3400 cm^−1^. 

### 2.8. Data Pre-Processing

All spectra were pre-processed using the PLS toolbox in MATLAB (MathWorks, Natick, MA, USA). The analysis was performed in the 1400 to 900 cm^−1^ region on the second derivative using Savitzky–Golay algorithm, polynomial order of 2 and 15 smoothing points then normalized using SNV and mean cantered on all technical replicates. 

### 2.9. Data Analysis

Second derivative spectra were compared following treatment with different drugs during dehydration and rehydration. Initially, the dataset was refined by excluding spectra that contained excessive amounts of noise due to water vapour, observed as sharp peaks between 1650 and 1500 cm^−1^ and spectra that were overwhelmed by water contributions, observed as a strong δOH at 1630 cm^−1^ and diminished or non-existent signal between 1400 and 900 cm^−1^. 

Principle Component Analysis (PCA) was performed on spectra of hydrated samples treated with the different drugs for ssDNA and dsDNA using MATLAB (Math Works, Natick, MA, USA). Outliers, either with high Hotelling T^2^ values and high Q-residuals or falling outside the 95% confidence interval, were also excluded from the analysis and PCA was performed on the finalised dataset.

## 3. Results

Complexes **PFB** and **TMB** were prepared by photo-induced substitution of pyridine by a carboxylate ion with concomitant protonation of one amide nitrogen (Figure 1). Both gave satisfactory microanalyses and appropriate mass spectra including (M+H)^+^ ions with the expected isotope patterns. Both ^1^H and ^19^FNMR spectra provided evidence of two different 2,3,5,6-tetrafluorophenyl groups. The NH resonance was clear for **PFB** but was overlapped by H2,6 (py) for **TMB**. Signals for or overlapping the NH resonance gave lower than expected integrations, possibly owing to partial exchange with water in the solvent. As the protonated nitrogen atom is chiral (H, CH_2_, p-HC_6_F_4_, Pt substituents), the adjacent CH_2_ group is prochiral and gives two resonances separated by ca 1ppm. The ^3^J_H,Pt_ coupling constants of the pyridine ligand (39 Hz) are larger than that of Pt103 (34 Hz) [29] and those of the [Pt{N(C_6_F_4_X-4)CH_2_}_2_(py)_2_] (X = F, Cl, Br, I, Me) complexes (33–35 Hz) where pyridine is trans to a tetrafluorophenylamide nitrogen but are similar to those of [Pt(py)_2_(H_2_NCH_2_)_2_](O_2_CC_6_F_5_)_2_] (39 Hz) [29] and [Pt{N(*p*-XC_6_F_4_)(CH_2_)_2_NMe_2_}(py)(Y)] (Y = halide; X = H, F, Cl, Br, Me) (39 Hz) [31] where pyridine is trans to an amine nitrogen. Accordingly, the pyridine ligand in **PFB** and **TMB** is trans to the amine (NH) nitrogen as in Figure 1.

The drugs were tested in vitro and shown to be effective against leukaemia cell lines (L1210, L1210/DDP [32]）.

FTIR spectra of the synthesized drugs **PFB** and **TMB** were recorded in the spectral region from 1800 to 900 cm^−1^ (Figure 3). 

The ν_as_(COO^−^) shows an intense band between 1658–1617 cm^−1^ while the ν_s_(COO^−^) appears between 1363–1341 cm^−1^. The ν(C-F) shows two bands at ~934 and 990 cm^−1^, the latter resulting from vibrations characteristic of the pentafluorobenzoate group. The bands at approximately 1495, 1047 and 1500 cm^−1^ are also characteristic of the pentafluorobenzoate group vibrations.

ATR-FTIR spectra were recorded of all DNA-drug samples in the spectral region from 1800–850 cm^−1^ where the DNA phosphodiester bands are located. Table 1 shows the DNA band assignment in this region. Raw spectra are shown in Appendix A. Figure 4 shows the second derivative spectra during dehydration (left) and rehydration (right) from DNA treated with acetone/water or saline as the controls along with compounds **PFB**, **TMB** and cisplatin. In a previous study the dehydration and rehydration effects on the DNA conformation were especially noticeable after subtraction of water bands [33]. Here, each set of spectra from dehydrated and rehydrated mixtures for both the control and the treated DNA were pre-processed separately. The raw infrared spectra of ssDNA and dsDNA recorded during rehydration are shown in Figure 4. The dsDNA spectra, show an increase in the 1083 and 1054 cm^−1^ bands and an intense band at 1711 cm^−1^ going from the A-DNA to B-DNA conformation. ssDNA shows similar changes upon rehydration. The return to the B-DNA conformation can be used as a measure to test whether drugs have interacted with the DNA. If drugs bind strongly to the DNA then upon rehydration the DNA will not return to the B-DNA conformation.

To monitor the effect of hydration on DNA conformation following incubation with the drugs, all samples were measured in the hydrated state, during drying and after rehydration. Figure 5 shows second derivative spectra of control DNA and drug-treated DNA in the hydrated, dehydrated and rehydrated state. Cisplatin also serves as a control because its mechanism of interaction with DNA is well established [37]. First, dsDNA was treated with cisplatin, dried on the ATR crystal and rehydrated with water. The same process of dehydration and rehydration was performed on another sample of DNA mixed only with saline. The cisplatin-induced effects on DNA after rehydration are highlighted in the spectra shown in Figure 5. The most affected bands were at 1716, 1225, 1088, 1051 and 968 cm^−1^. The bands in the rehydrated cisplatin-treated DNA show similar changes to those that occurred during the dehydration process. This indicates the damage in the DNA was influenced by the interaction with cisplatin.

In the controls, a noticeable change in the spectra of the two bands around 1088 and 1051 cm^−1^ is observed during dehydration. They both decreased in intensity relative to the 1051 cm^−1^ band. While the band at 1088 cm^−1^ is more intense in the hydrated state, the band at 1051 cm^−1^ is more intense in the dehydrated state. According to the literature this indicates that the DNA conformation has changed from B-DNA to A-DNA [34]. This change is only related to the deficiency in water content and is a temporary change that returns back to the B-conformation upon rehydration. Therefore in this case the DNA conformation is not affected by the acetone water mixture. The band at 1225 cm^−1^ shifts and splits during dehydration and appears as two bands at 1234 and 1215 cm^−1^ in the dehydrated A-DNA. The band at 968 cm^−1^ increases in intensity and shifts towards 966 cm^−1^, while the band at 1716 cm^−1^ shifts towards 1711 cm^−1^. Whereas the acetone treated A-DNA returns to the B-form upon rehydration as evidenced by the fact that 1088 cm^−1^ and 1051 cm^−1^ show similar changes to the acetone: DNA mixture during dehydration and rehydration.

Drugs **PFB** and **TMB** show a mixture of B-DNA and A-DNA in the hydrated state as evinced by the split of the band at 1225 cm^−1^. This band is assigned to the PO_2_^−^ asymmetric-stretching vibration and indicates that the mode of interaction for these drugs is similar. In the dehydrated state for the same drugs the 1225 cm^−1^ had split into two bands 1215 and 1234 cm^−1^, which indicates the partial transformation into the A-DNA conformation [38]. The PO_2_^−^ band at 1225 cm^−1^ in the control spectra in the hydrated state indicates the exclusive presence of B-DNA, while in the dehydrated the A-DNA conformation dominates the spectral profile, indicated by the blue shift to 1234 cm^−1^. The band 1051 cm^−1^ shows a decrease in intensity and a red shift towards 1060 cm^−1^ for all the drugs compared to the control DNA where there is no effect on this band.

Another feature in characteristic DNA conformation change is the band at 968 cm^−1^. In the hydrated state of the control B-DNA this band is at 968 cm^−1^, while it is slightly red-shifted towards 966 cm^−1^ in the dehydrated state. In the dehydrated state of compounds, **TMB** and cisplatin there is a decrease in intensity and a red shift towards 966 cm^−1^ while it decreases in intensity for PFB. In rehydration, the band keeps the same situation as in the dehydration state for compounds **PFB**, **TMB** and cisplatin.

The base pairing mode of dsDNA at 1716 cm^−1^ in the control DNA maintains the intensity and the position in both, the hydrated and the dehydrated state, while it shows a clear change in the treated DNA with the drugs. For dsDNA treated with **PFB** and cisplatin a significant decrease in intensity and a red-shift from 1716 to 1701–1711 cm^−1^ is visible in the hydrated state. That change is also observed in the dehydrated state. This indicates a partial separation of the A-DNA double helix into single strands with weaker base pairing interactions [26].

An intensity change of the symmetric phosphate stretching at 1088 cm^−1^ is also associated with a transformation from B-DNA to A-DNA. In the control DNA, there is no change in this band in the hydrated state but a slight decrease in the band intensity at 1088 cm^−1^ in respect to the band at 1051 cm^−1^ in the dehydrated state. The band decreased in intensity significantly in the hydrated state in spectra of dsDNA treated with **TMB**, while only a slight decrease in intensity is visible in spectra of dsDNA treated with **PFB**.

### 3.1. Interaction of Drugs with Single Stranded DNA

To initiate cell proliferation, DNA is replicated. In the course of DNA-replication, dsDNA is unwound into ssDNA. As tumour cells have an increased proliferation rate, more ssDNA is present compared to healthy cells. Targeting ssDNA therefore increases the selectivity of an anticancer drug towards tumour cells. Consequently, we are particularly interested in the interaction of the studied drugs with ssDNA. Similar to the interaction studies with dsDNA, ssDNA was treated with the drugs in the hydrated state and the drug-DNA interaction was monitored with IR spectroscopy during dehydration and rehydration. The 2nd derivative IR spectra of ssDNA treated with **PFB**, **TMB** and cisplatin during dehydration and rehydration are shown in Figure 6.

Spectra of ssDNA treated with **PFB** and **TMB** show more intense A-DNA conformation indicators compared to dsDNA and ssDNA treated with **PFBcisplatin**. The asymmetric phosphodiester band at 1225 cm^−1^ in the hydrated state is shifted towards 1234 cm^−1^ indicating that **PFB** has stronger interactions with ssDNA compared to dsDNA. Thus, **PFB** might be more selective towards tumour cells compared to healthy cells.

The bands observed in both ssDNA and dsDNA spectra interacted with **PFB** and **TMB** between 1450 and 1525 cm^−1^ are from the tetrafluorophenylgroup in the drugs as discussed above.

The water content played a role in keeping the B-like DNA conformation for ssDNA for some time during dehydration in the controls. In ssDNA treated with **PFB**, **TMB** and cisplatin, the changes in DNA conformation start to appear while the samples are still in the hydrated state.

### 3.2. Principle Component Analysis

In order to further investigate the effects of **PFB** and **TMB** versus the controls, PCA was applied on the second derivative in the spectral region 1400–900 cm^−1^. For each drug there were three trials and multiple technical replicates for each trail. Cisplatin was not included in the PCA because it was solvated in saline while compounds **PFB** and **TMB** were in acetone thus were inappropriate to include in the same model. Figure 7A shows the PC1 versus PC2 scores plot depicting ssDNA treated with **PFB** and **TMB** and the non-treated DNA (control in acetone/water) in the dehydrated state, while Figure 7B shows the corresponding scores plot for the rehydrated ssDNA. Figure 7C shows the analogous scores plot for the dehydrated dsDNA, while Figure 7D shows the scores plot for the rehydrated dsDNA with and without drug treatment. In all cases, for ssDNA and dsDNA both in the hydrated and dehydrated state, the **TMB** clusters closely to the DNA/acetone/water control whereas **PFB** is considerably separated along PC1. The spread observed along PC2 is from different levels of hydration between the technical replicates.

Figure 8A shows the loadings plot for the rehydrated ssDNA, while Figure 8B shows the loadings plot for the dehydrated ssDNA. The positive score values are correlated with the negative loadings values because the PCA was performed on the second derivative spectra. The band at 1084 cm^−1^ is assigned to the $\nu_{s}(PO_{2}^{-})$ from the DNA phosphodiester backbone and appears as a very strong loading associated with the control ssDNA and is absent in the negative loadings associated with the drug incubated ssDNA spectra. The 974 cm^−1^ band also appears as a strong PC1 loading in the ssDNA control and is assigned to the dianionic phosphodiester vibration of DNA. This loading is not observed in the drug treated controls and demonstrates that both compounds **PFB** and **TMB** significantly disrupt the DNA phosphodiester backbone. Figure 8C,D shows the loadings plots for the rehydrated and dehydrated dsDNA. In this case the negative loadings are associated with the controls and once again show a strong $\nu_{s}(PO_{2}^{-})$ loading at 1084 cm^−1^ and also a strong band at 974 cm^−1^, which is absent in the drug incubated dsDNA. The results confirm that the drugs are substantially disrupting the phosphodiester backbone in the dsDNA in a similar way to the ssDNA as previously shown for Pt103 and cisplatin in cells [26]. In summary the transition from B-DNA to A-DNA is reversible when rehydrating the untreated control DNA but it is irreversible after treating DNA with the drugs especially **PFB**, which has a more dramatic effect on DNA than **TMB**. When comparing the vibrational modes of nucleobases with the vibrational modes of the backbone, they have lower intensity and it is not possible to track the changes in vibrations of nucleobases or determine if there is any type of binding to the drugs via the DNA bases at this stage.

## 4. Conclusions

The study shows how infrared spectroscopy can be used to study the interaction between some platinum drugs and DNA. It was found that dsDNA and ssDNA treated with PFB and TMB transform from B-DNA to A-DNA during dehydration and do not return back to the B-DNA conformation upon rehydration. Untreated control DNA in acetone/water, on the other hand, transforms from B-DNA to A-DNA during dehydration but is able to return back to its original conformation upon rehydration. The spectroscopic results indicate that the platinum complexes have a similar effect to cisplatin. This indicates a similar mechanism of interaction with DNA. The ability of infrared spectroscopy to study conformational dynamics opens up a new pathway to explore DNA:drug interactions with the potential to screen for new therapeutic agents.

## Figures and Tables

**Figure 1 sensors-18-04297-f001:**
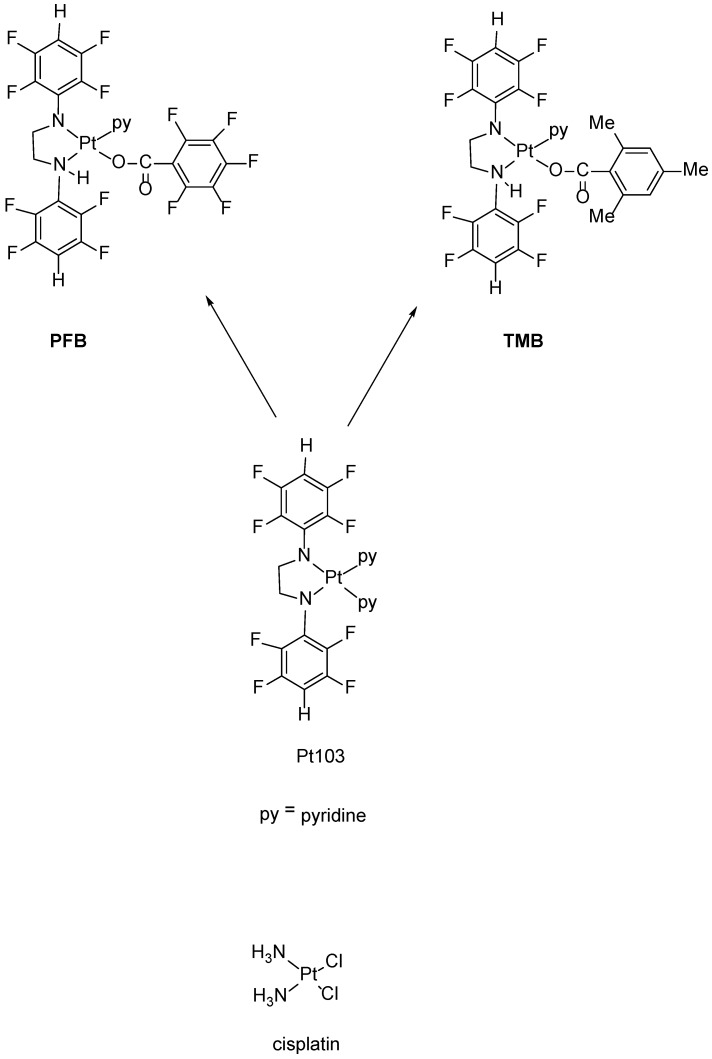
Chemical structures of **PFB**, **TMB** and cisplatin.

**Figure 2 sensors-18-04297-f002:**
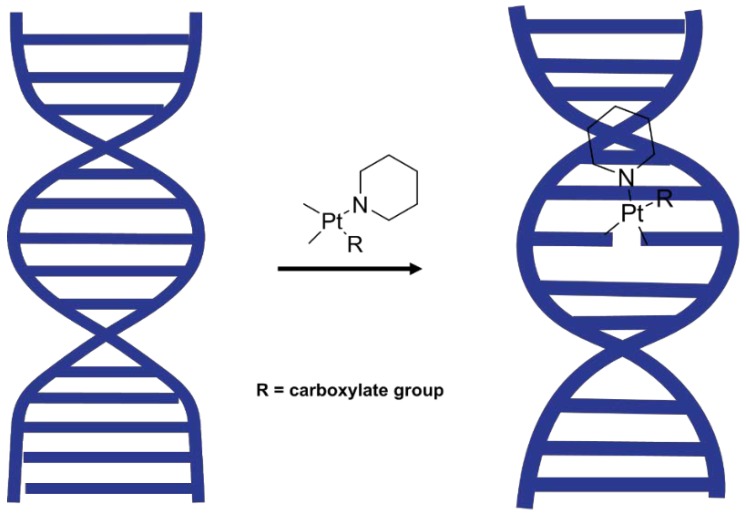
DNA interaction with Pt^II^ and conformation change.

**Figure 3 sensors-18-04297-f003:**
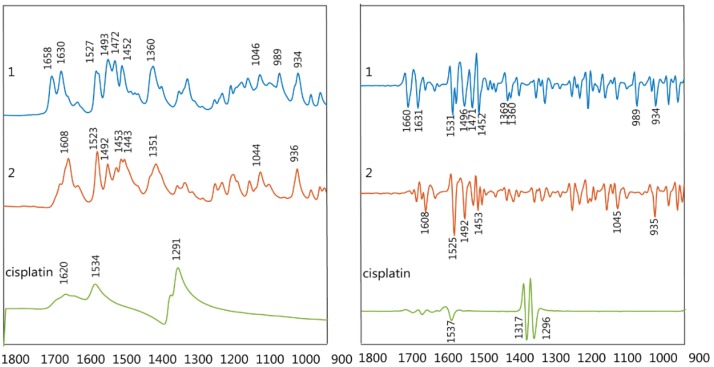
Infrared (IR) raw spectra (**left**) and 2nd derivative (**right**) of compounds **PFB** (1), **TMB** (2) and cisplatin.

**Figure 4 sensors-18-04297-f004:**
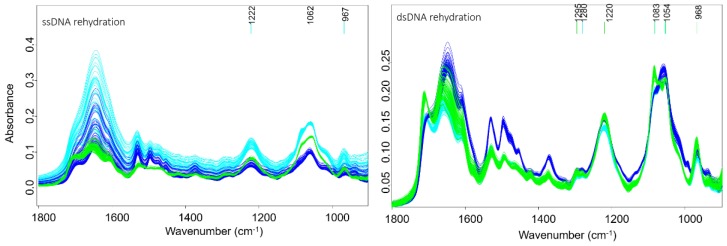
Raw IR spectra of the transition of ss and ds A-DNA to B-DNA upon rehydration. Green: control (DNA treated with acetone), dark blue: DNA treated with PFB, light blue: DNA treated with TMB.

**Figure 5 sensors-18-04297-f005:**
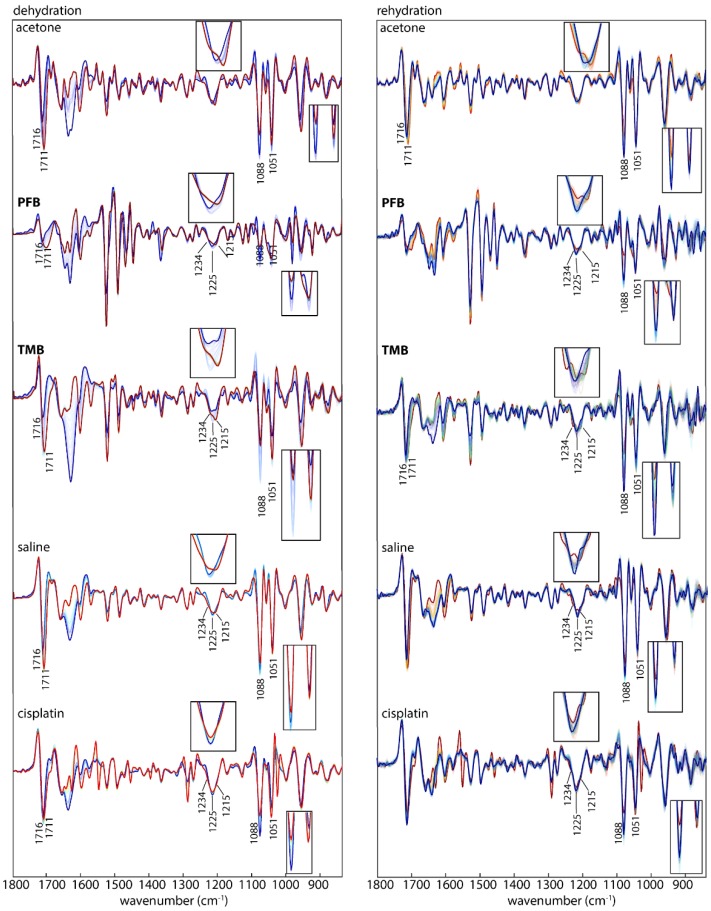
IR Average spectra (second derivative) of dsDNA treated with acetone (control 1) **PFB**, **TMB**, saline (control 2) and cisplatin in the course of dehydration (**left**) colour-coded from blue (hydrated) to red (dehydrated) and in the course of rehydration (**right**) colour-coded from red (dehydrated) to blue (hydrated). The inserts show spectral features around 1225 and 1088/1051 cm^−1^.

**Figure 6 sensors-18-04297-f006:**
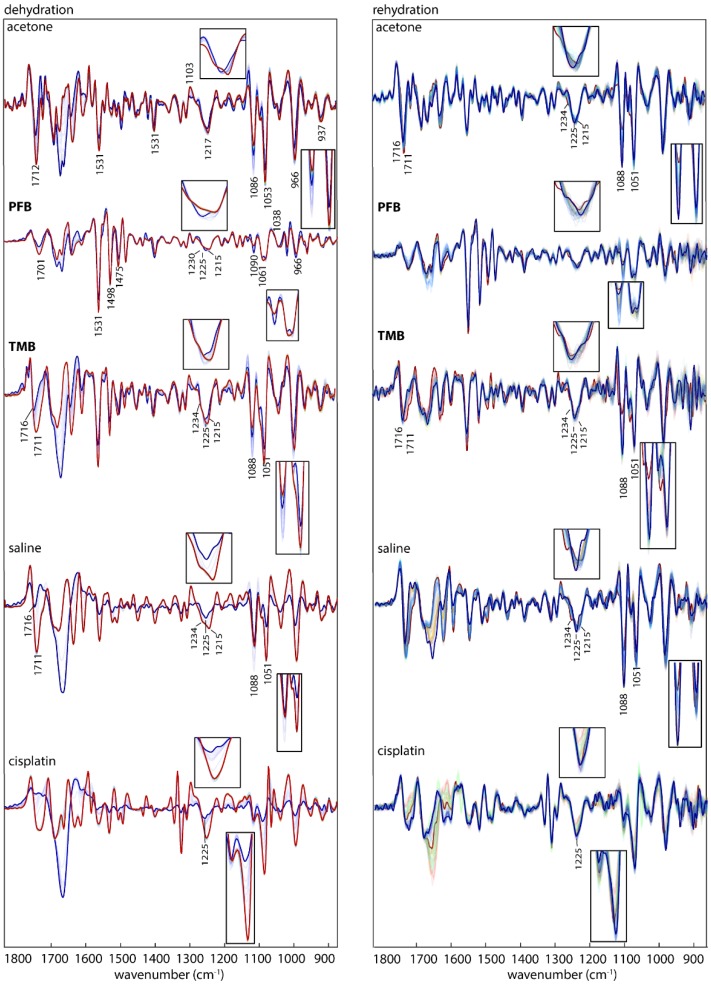
IR Average spectra (second derivative) of ssDNA treated with acetone (control 1) **PFB**, **TMB**, saline (control 2) and cisplatin in the course of dehydration (**left**) colour-coded from blue (hydrated) to red (dehydrated) and in the course of rehydration (**right**) colour-coded from red (dehydrated) to blue (hydrated). The inserts show spectral features around 1225 and 1088/1051 cm^−1^.

**Figure 7 sensors-18-04297-f007:**
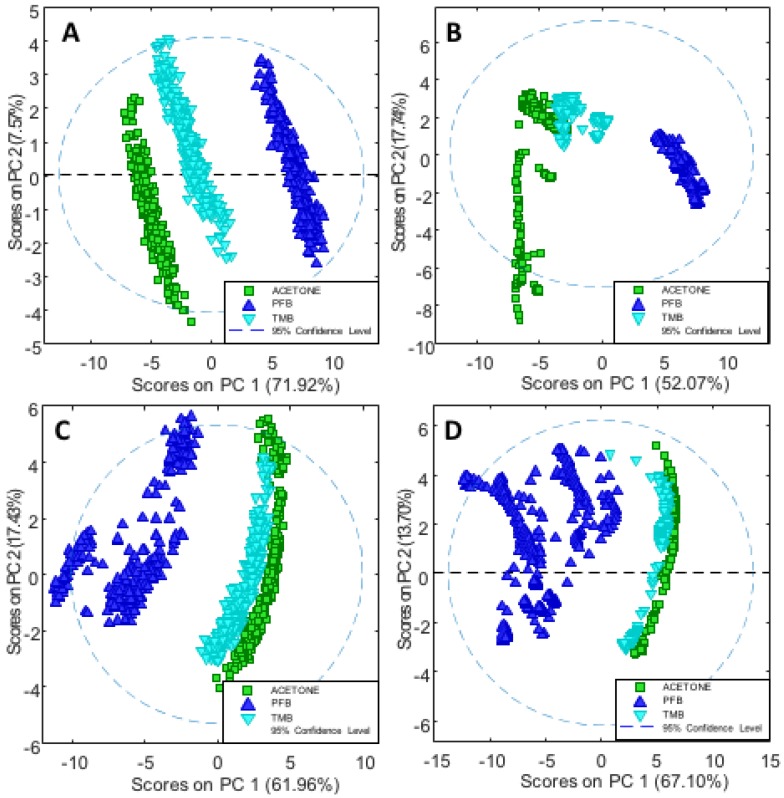
(**A**) PC1 versus PC2 scores plot depicting ssDNA treated with the **PFB** and **TMB** and the non-treated DNA (control in acetone/water) in the dehydrated state; (**B**) PC1 versus PC2 scores plot for the rehydrated ssDNA with and without drug treatment; (**C**) PC1 versus PC2 scores plot for dehydrated dsDNA with and without drug treatment; (**D**) PC1 versus PC2 Scores Plot for rehydrated dsDNA with and without the drug treatment.

**Figure 8 sensors-18-04297-f008:**
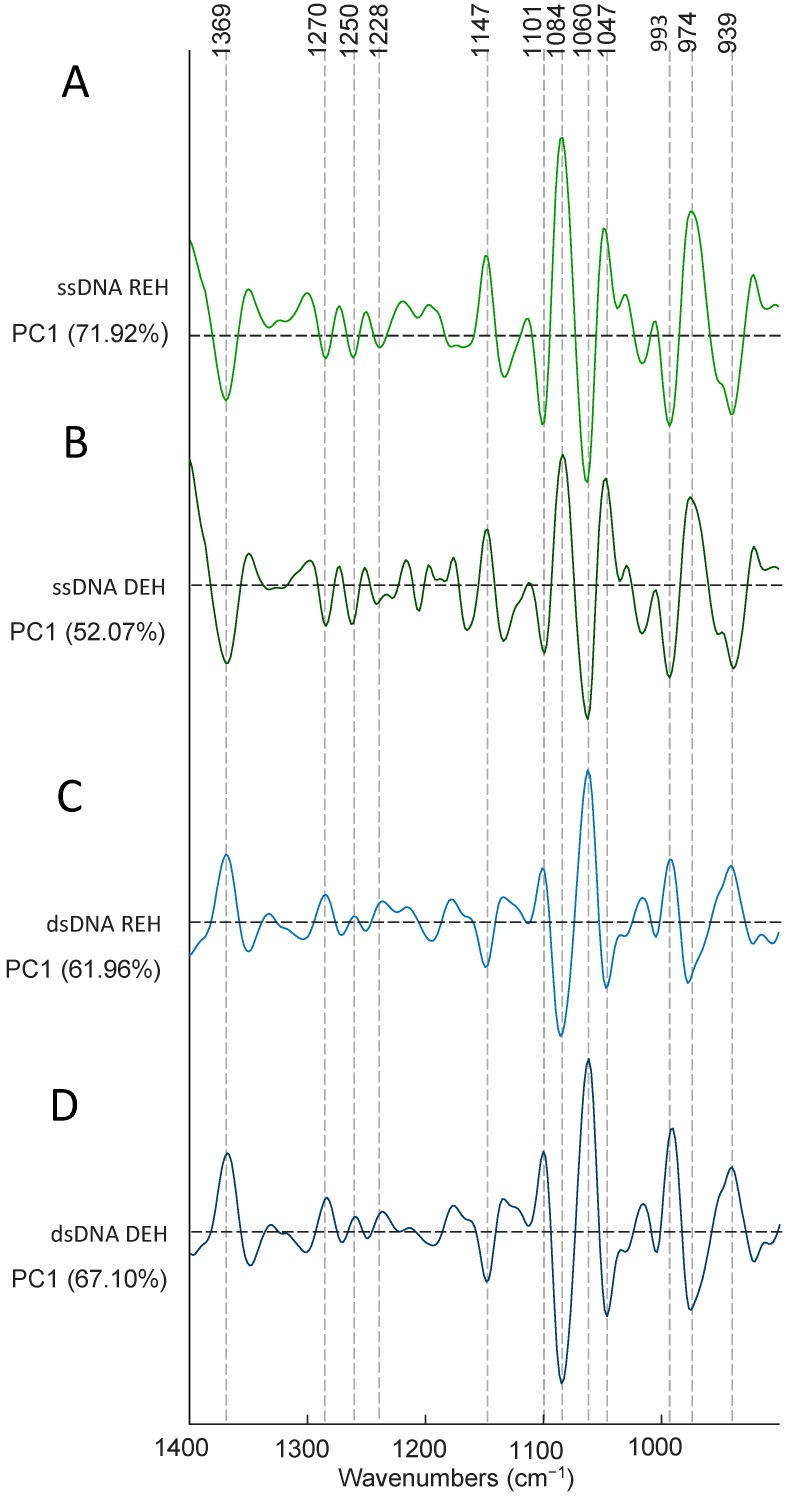
(**A**) PC1 loadings plot depicting ssDNA treated with **PFB** and **TMB** and the non-treated DNA (control in acetone/water) in the dehydrated state. The positive loadings are associated with the controls and the negative loadings are associated with the drug inoculated cells; (**B**) PC1 loadings plot for the rehydrated ssDNA with and without drug treatment. The positive loadings are associated with the negative scores. The positive loadings are associated with the controls and the negative loadings are associated with the drug inoculated cells; (**C**) PC1 loadings plot for dehydrated dsDNA with and without drugs. The positive loadings are associated with the drug inoculated cells and the negative loadings are associated with the control cells; (**D**) PC1 versus PC2 loadings plot for rehydrated dsDNA with and without the drugs. The positive loadings are associated with the drug inoculated cells and the negative loadings are associated with the control cells.

**Table 1 sensors-18-04297-t001:** Peak positions for DNA conformational bands.

A Conformation (cm^−1^)	B Conformation (cm^−1^)	Assignment
1705	1712	Base pair carbonyl V(C=O) [34,35]
1418	1422	C2/C3′-endo deoxyribose [36]
1275	1281	Unidentified
1238	1225	Asymmetric phosphate stretching [35]
1188	Absent	C3′-endo-sugar phosphate [35]
1088	1088 *	Symmetric phosphate [35]
1055	1055 *	Backbone v(C-O) [35]
968	970	Backbone v(C-C) [35]

* Indicates loss of intensity.

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
