# Peer review of "The Application of ATR-FTIR Spectroscopy and the Reversible DNA Conformation as a Sensor to Test the Effectiveness of Platinum(II) Anticancer Drugs"

_sensors, 2018, doi:10.3390/s18124297_

Reviewer 1 Report

This manuscript shows a very innovative way of using infrared spectroscopy to monitor changes in conformation of DNA. In particular, it studies the interaction with some platinum drugs with DNA. The importance of this study is well grounded in the introduction, and it is of great importance to understanding the effect of platinum drugs on DNA. I would recommend this paper for publication after reviewing the following observations.

For the synthesis of PFB (2.2.1) and TMB (2.2.2) would be useful to present the spectra in a supplementary figure.

In description of “Figure 2” in line 255 and 263 does not correspond to Figure 2. It seems it refers to figure 3, but still in description (263) refers to left and right in a mistaken way. On the other hand, the actual Figure 2, is not mentioned anywhere else in the manuscript. This should be corrected.

In the figure 3, 4 and 5 caption it is not clear what 1 and 2 stands for. It would be more helpful to the reader to display them as PFB and TMB.

Line 127 closing parenthesis and extra blank space

Line 129 blank space

Line 280 typo “serves”

Line 294 cm-1

Line 303 typo “acetone”

In figure 4 and 5, it would be useful to show the zoom in also of bands 1088 and 1051, since it is discussed thoroughly through the article.

Line 309 typo “.”

Line 317 “decreases”

Line 318 “keeps”

Line 339 Instead of 1,2 use its abbreviations

Line 390 “blank space missing”

Line 395 extra blank space

Line 407 “It” instead of “In”

Figure S1 is not mentioned in the text.

Author Response

Reviewer 1

Author Response

Reviewer 1

For the synthesis of PFB (2.2.1) and TMB (2.2.2) would be useful to present the spectra in a supplementary figure.

We added the IR, 1HNMR, 19FNMR and ESMS spectra in the SI and mention them in the main text.

In description of “Figure 2” in line 255 and 263 does not correspond to Figure 2. It seems it refers to figure 3, but still in description (263) refers to left and right in a mistaken way. On the other hand, the actual Figure 2, is not mentioned anywhere else in the manuscript. This should be corrected.

We added the missing figure 4, corrected for the confusion in descriptions and mentioned figure 2 in the text.

In the figure 3, 4 and 5 caption it is not clear what 1 and 2 stands for. It would be more helpful to the reader to display them as PFB and TMB.

We replaced 1and 2 with PFB and TMB in the figure captions.

Line 127 closing parenthesis and extra blank space

Line 129 blank space

Line 280 typo “serves”

Line 294 cm-1

Line 303 typo “acetone”

Line 309 typo “.”

Line 317 “decreases”

Line 318 “keeps”

Line 339 Instead of 1,2 use its abbreviations

Line 390 “blank space missing”

Line 395 extra blank space

Line 407 “It” instead of “In”

We corrected for these typos and formal mistakes.

In figure 4 and 5, it would be useful to show the zoom in also of bands 1088 and 1051, since it is discussed thoroughly through the article.

We added an insert with a zoom of these bands in the figures.

Figure S1 is not mentioned in the text.

We mention Figure S1 and the additional figures from the SI in the main text.

Reviewer 2 Report

The authors presented the application of ATR-FTIR to study the interaction between DNA and Platinum.

The work is well concieved, the results are well described, with thw appropriate types of figures.

For me it can be published in the present form

Author Response

Reviewer 2

No comments from reviewer 2 needed to be addressed.

Reviewer 3 Report

The paper by Al-Jorani et al. concerns the use of ATR-FTIR spectroscopy as a sensor for testing the effectiveness of some of the novel Pt(II) cancer drugs. The manuscript is clearly written, and the scientific methodology seems to be sound. The Authors support their careful studies with a variety of modern spectroscopic methods. However, before considering the work for publication, the Referee has some questions/suggestions.

1) In general, the Authors were mixing the drug with the solutions of DNA (ssDNA and dsDNA). Therefore we could expect that the spectra showing the changes in DNA could be also “contaminated” with the drugs, especially in the vicinity of the bands that were also present in ATR-FTIR spectra of the drugs. Can the Authors show/comment on the comparison of raw ATR-FTIR spectra of the drugs, ssDNA and dsDNA upon reaction (only second order derivative spectra are actually shown)?

2)  When analysing the (re)hydrated media, the Referee understands that there was “water” contamination in the system. Can the Authors explain if the water contribution has been an issue? If so, what was the measure the Authors have undertaken to correct against its contribution?

3) Can the Authors explain why the results (concerning the course of the reaction with the drugs) were presented in the form of second order derivative spectra rather than raw spectra?

4) For considering a method as a sensor, one has to consider (measure) or show its sensitivity to probe a specific reaction. In the paper, there is not any statistics (testing) employed to show the strength of the differences between various stages of the reaction, for example,  by computing the average of the heights/areas of some of the most prominent bands. This issue is practically important as, with the current data presentation, the spectral differences in second order derivative spectra are very subtle which prevents from judging their statistical significance, though the PCA shows global differences – but not concerning (numerically) the strength of differences between single bands. In particular, it would be beneficial for the paper to visualize the transition between A-DNA and B-DNA (the key remark that follows from the paper) using spectral signatures (height/area/position) of their marker bands (i.e. marker vs time or the measurement number). The Authors wrote “….Both gave satisfactory microanalyses and appropriate mass spectra including parent ions with the expected isotope patterns...“. What the does word “satisfactory” mean? What was the measure of their spectral reproducibility?

5) How was the rehydratation process monitored and judged?

6) The Authors wrote: “...The analysis was performed in the 1400-900 cm-1 region on the second derivative using Savitsky-Golay algorithm, polynomial order of 2, and 15 smoothing points then normalized using SNV and mean cantered...”. Why did the Authors perform the mean centering after SNV? Upon SNV, the mean is actually 0 (mu=0).

7) The Authors should check the references to all the Figures in the text. For example, Fig 2 does not show "the second order derivative spectra", as so does Fig 4 (please see the line 263).

8) Over the text, some of the sentences are repeated. For example, lines 263-264, 268-269, 278-280 repeated the same information.

Author Response

Reviewer 3

In general, the Authors were mixing the drug with the solutions of DNA (ssDNA and dsDNA). Therefore we could expect that the spectra showing the changes in DNA could be also “contaminated” with the drugs, especially in the vicinity of the bands that were also present in ATR-FTIR spectra of the drugs. Can the Authors show/comment on the comparison of raw ATR-FTIR spectra of the drugs, ssDNA and dsDNA upon reaction (only second order derivative spectra are actually shown)?

There might be possible contaminations with drugs. However, the major bands in the PCA loadings plot are nucleic acid, lipid and protein modes. Only very minor contributions from drugs are visible. These are not major contributors to the loadings plot from drug bands and hence they are not important for the separation observed in the PCA scores plot.

A figure of raw spectra of DNA during dehydration was added.

When analysing the (re)hydrated media, the Referee understands that there was “water” contamination in the system. Can the Authors explain if the water contribution has been an issue? If so, what was the measure the Authors have undertaken to correct against its contribution?

Using ATR-FTIR spectroscopy, the penetration depth of the light beam is only a few microns. So the water contribution did not swamp the spectral features. The spectral features from water do not interfere with the spectral features from the analyte. Furthermore, only spectral regions without major water bands were chosen for PCA.

Can the Authors explain why the results (concerning the course of the reaction with the drugs) were presented in the form of second order derivative spectra rather than raw spectra?

By calculating the second derivative, hidden spectral features can be resolved as for example shoulders in the raw spectra appear as distinct bands in the second derivative spectra and baseline variation is minimised.

For considering a method as a sensor, one has to consider (measure) or show its sensitivity to probe a specific reaction. In the paper, there is not any statistics (testing) employed to show the strength of the differences between various stages of the reaction, for example,  by computing the average of the heights/areas of some of the most prominent bands. This issue is practically important as, with the current data presentation, the spectral differences in second order derivative spectra are very subtle which prevents from judging their statistical significance, though the PCA shows global differences – but not concerning (numerically) the strength of differences between single bands. In particular, it would be beneficial for the paper to visualize the transition between A-DNA and B-DNA (the key remark that follows from the paper) using spectral signatures (height/area/position) of their marker bands (i.e. marker vs time or the measurement number).

It is a fair point raised by the reviewer and we could use averages of particular DNA bands to infer conformational DNA distortion. However, we consider that PCA is doing much the same thing not on a global scale but rather on the major bands varying in the spectra i.e. the DNA bands. We can see that there are a number of dominant bands in the drug treated spectra that show a reduction in the overall intensity compared to the control DNA that dominate the loadings plot. There are also indications that the DNA conformation in the drug incubated experiments developed an A-DNA conformation upon rehydration.  The separation along the PC1 axis from the centroid of the drug inoculated DNA cluster to the centroid of control DNA DNA represents the measure of DNA distortion/denaturation and can be used as a sensor. We have now included a Figure showing rehydration of ssDNA and dsDNA showing the A-DNA to B-DNA conformation as requested.

The Authors wrote “….Both gave satisfactory microanalyses and appropriate mass spectra including parent ions with the expected isotope patterns...“. What the does word “satisfactory” mean? What was the measure of their spectral reproducibility?

Satisfactory means “acceptable, adequate, or sufficient.”  In our context it means adequate to support the proposed composition and also indicates acceptable purity.  The results also meet the criteria for satisfactory microanalyses for organic compounds viz not outside +/- 0.3% of the calculated values- a stringent standard when applied to metal-organic compounds. The 19F and 1H NMR and IR spectra were reproducible both for the one investigator and between two investigators.

How was the rehydration process monitored and judged?

The rehydration process was monitored by observing the intensity increase in the OH- stretching mode at 3400 cm-1.

The Authors wrote: “...The analysis was performed in the 1400-900 cm-1 region on the second derivative using Savitsky-Golay algorithm, polynomial order of 2, and 15 smoothing points then normalized using SNV and mean cantered...”. Why did the Authors perform the mean centering after SNV? Upon SNV, the mean is actually 0 (mu=0).

We agree with the reviewer it probably does not make any difference and you will still get the same loadings and separation by using SNV alone but mean centering is a default setting when performing PCA.

The Authors should check the references to all the Figures in the text. For example, Fig 2 does not show "the second order derivative spectra", as so does Fig 4 (please see the line 263).

The confusing naming of the figures was corrected.

Over the text, some of the sentences are repeated. For example, lines 263-264, 268-269, 278-280 repeated the same information.

We reread the paper removed unnecessary repetition.